# Finite Sample Analysis Of Dynamic Regression Parameter Learning

**Mark Kozdoba**
Technion, Israel Institute of Technology
markk@ef.technion.ac.il

**Edward Moroshko**
Technion, Israel Institute of Technology
edward.moroshko@gmail.com

**Shie Mannor**
Technion, Israel Institute of Technology and NVIDIA Research
shie@ee.technion.ac.il

**Koby Crammer**
Technion, Israel Institute of Technology
koby@ee.technion.ac.il

## Abstract

We consider the dynamic linear regression problem, where the predictor vector may vary with time. This problem can be modeled as a linear dynamical system, with non-constant observation operator, where the parameters that need to be learned are the variance of both the process noise and the observation noise. While variance estimation for dynamic regression is a natural problem, with a variety of applications, existing approaches to this problem either lack guarantees altogether, or only have asymptotic guarantees without explicit rates. In particular, existing literature does not provide any clues to the following fundamental question: In terms of data characteristics, what does the convergence rate depend on? In this paper we study the global system operator – the operator that maps the noise vectors to the output. We obtain estimates on its spectrum, and as a result derive the first known variance estimators with finite sample complexity guarantees. The proposed bounds depend on the shape of a certain spectrum related to the system operator, and thus provide the first known explicit geometric parameter of the data that can be used to bound estimation errors. In addition, the results hold for arbitrary sub Gaussian distributions of noise terms. We evaluate the approach on synthetic and real-world benchmarks.

## 1 Introduction

A dynamic linear regression (West and Harrison, 1997, Chapter 3), or non-stationary regression, is a situation where we are given a sequence of scalar *observations* $\{Y_t\}_{t \leq T} \subset \mathbb{R}$, and *observation vectors* $\{u_t\}_{t \leq T} \subset \mathbb{R}^n$ such that $Y_t = \langle X_t, u_t \rangle + z_t$ where $X_t \in \mathbb{R}^n$ is a regressor vector, and $z_t$ a random noise term. In contrast to a standard linear regression, the vector $X_t$ may change with time. One common objective for this problem is at time $T$, to estimate the trajectory of $X_t$ for $t \leq T$, given the observation vectors and observations, $\{u_t\}_{t \leq T}$, $\{Y_t\}_{t \leq T}$, and possibly to forecast $Y_{T+1}$ if $u_{T+1}$ is also known.

36th Conference on Neural Information Processing Systems (NeurIPS 2022).

In this paper we model the problem as follows:

$$X_{t+1} = X_t + h_t \tag{1}$$

$$Y_t = \langle X_t, u_t \rangle + z_t, \tag{2}$$

where $\langle \cdot, \cdot \rangle$ is the standard inner product on $\mathbb{R}^n$, $z_t$, the *observation noise*, are zero-mean sub Gaussian random variables, with variance $\eta^2$, and the *process noise* variables $h_t$ take values in $\mathbb{R}^n$, such that coordinates of $h_t$ are zero-mean sub Gaussian, independent, and have variance $\sigma^2$. All $h_t$ and $z_t$ variables are assumed to be mutually independent. The vectors $u_t$ are an *arbitrary* sequence in $\mathbb{R}^n$, and the observed, known, quantities at time $T$ are $\{Y_t\}_{t \leq T}$ and $\{u_t\}_{t \leq T}$.

The system (1)-(2) is a special case of a Linear Dynamical System (LDS). As is well known, when the parameters $\sigma, \eta$ are given, the mean-squared loss optimal forecast for $Y_{T+1}$ and estimate for $X_T$ are obtained by the Kalman Filter (Anderson and Moore, 1979; Hamilton, 1994; Chui and Chen, 2017). In this paper we are concerned with estimators for $\sigma, \eta$, and *finite* sample complexity guarantees for these estimators.

Let us first make a few remarks about the particular system (1)-(2). First, as a natural model of time varying regression, this system is useful in a considerable variety of applications. We refer to West and Harrison (1997), Chapter 3, for numerous examples. In addition, an application to electricity consumption time series as a function of the temperature is provided in the experiments section of this paper. Second, one may regard the problem of estimating $\sigma, \eta$ in (1)-(2) as a pure case of finding the optimal *learning rate* for $X_t$. Indeed, the Kalman filter equations for (1)-(2), are given by (3)-(4) below, where (3) describes the filtered covariance update and (4) the filtered state update. Here $\bar{x}_t$ is the estimated state, given the observations $Y_1, \ldots, Y_t$, see West and Harrison (1997).

$$C_{t+1} = \frac{\eta^2}{\langle (C_t + \sigma^2 I) u_{t+1}, u_{t+1} \rangle + \eta^2} \left( C_t + \sigma^2 I \right) \tag{3}$$

$$\bar{x}_{t+1} = \bar{x}_t + \frac{C_{t+1}}{\eta^2} u_{t+1} \cdot (Y_{t+1} - \langle \bar{x}_t, u_{t+1} \rangle). \tag{4}$$

In particular, following (4), the role of $\sigma$ and $\eta$ may be interpreted as regulating how much the estimate of $\bar{x}_{t+1}$ is influenced, via the operator $\frac{C_{t+1}}{\eta^2}$, by the most recent observation and input $Y_{t+1}, u_{t+1}$. Roughly speaking, higher values of $\sigma$ or lower values of $\eta$ would imply that the past observations are given less weight, and result in an overfit of the forecast to the most recent observation. On the other hand, very low $\sigma$ or high $\eta$ would make the problem similar to the standard linear regression, where all observations are given equal weight, and result in a *lag* of the forecast. See Figure 3 in Supplementary Material Section A for an illustration.

Finally, it is worth mentioning that the system (1)-(2) is closely related to the study of *online gradient* (OG) methods (Zinkevich, 2003; Hazan, 2016). In this field, assuming quadratic cost, one considers the update

$$\bar{x}_{t+1} = \bar{x}_t + \alpha \cdot u_{t+1} \cdot (Y_{t+1} - \langle \bar{x}_t, u_{t+1} \rangle), \tag{5}$$

where $\alpha$ is the learning rate, and studies the performance guarantees of the forecaster $\langle \bar{x}_t, u_{t+1} \rangle$. Compared to (4), the update (5) is simpler, and uses a scalar rate $\alpha$ instead of the input-dependent operator rate $C_{t+1}/\eta^2$ of the Kalman filter. However, due to the similarity, every domain of applicability of the OG methods is also a natural candidate for the model (1)-(2) and vice-versa. As an illustration, we compare the OG to Kalman filter based methods with learned $\sigma, \eta$ in the experiments section.

In this paper we introduce a new estimation algorithm for $\sigma, \eta$, termed STVE (Spectrum Thresholding Variance Estimator), and prove finite sample complexity bounds for it. In particular, our bounds are an explicit function of the parameters $T$ and $\{u_t\}_{t=1}^T$ for any finite $T$, and indicate that the estimation error decays roughly as $T^{-\frac{1}{2}}$, with high probability. To the best of our knowledge, these are the first bounds of this kind. As we discuss in detail in Section 2, most existing estimation methods for LDSs, such as subspace identification (van Overschee and de Moor, 1996; Qin, 2006), or improper learning (Anava et al., 2013; Hazan et al., 2017; Kozdoba et al., 2019), do not apply to the system (1)-(2), due to non-stationarity. On the other hand, the methods that do apply to (1)-(2) either lack guarantees, or have only asymptotic analysis which in addition relies strongly on Guassianity of the noises.

Moreover, our approach differs significantly from the existing methods. We show that the structure of equations (1)-(2) is closely related to, and inherits several important properties from, the classical discrete Laplacian operator on the line — leading to new arguments that have not been explored in the literature. In particular, we use this connection to show that an appropriate inversion of the system produce estimators that are concentrated enough so that $\sigma$ and $\eta$ may be recovered. The heart of the paper is the new definition of the estimators that exploits explicitly the shape of a certain data dependent operator, and the subsequent concentration analysis. In particular, this approach yields the first known *geometric* parameters of the data that can be used to bound convergence rates.

The rest of the paper is organized as follows: The related work is discussed in Section 2 and Section 3 contains the necessary definitions. In Section 4 we describe in general lines the methods and the main results of this paper. The technical estimates on certain operator spectra, that are critical to the analysis and may be of independent interest, are stated in Section 5. In Section 6 we present experimental results on synthetic and real data. Due to space constraints, while we outline the main arguments in the text, the full proofs are deferred to the Supplementary Material.

## 2 Literature

We refer to Chui and Chen (2017); Hamilton (1994); Anderson and Moore (1979); Shumway and Stoffer (2011) for a general background on LDSs, the Kalman Filter and maximum likelihood estimation.

Existing approaches to the variance estimation problem may be divided into three categories: (i) General methods for parameter identification in LDS, either via maximum likelihood estimation (MLE) (Hamilton, 1994), or via subspace identification (van Overschee and de Moor, 1996; Qin, 2006). In particular, finite sample bounds for system identification were given in (Campi and Weyer, 2005; Vidyasagar and Karandikar, 2006) and in the recent work Tsiamis and Pappas (2019). (ii) Methods designed specifically to learn the noise parameters of the system, developed primarily in the control theory community, in particular via the innovation auto-correlation function, such as the classical Mehra (1970); Belanger (1974), or for instance more recent Wang et al. (2017); Dunik et al. (2018). (iii) *Improper Learning* methods, such as Anava et al. (2013); Hazan et al. (2017); Kozdoba et al. (2019). In these approaches, one does not learn the LDS directly, but instead learns a model from a certain auxiliary class and shows that this auxillary model produces forecasts that are as good as the forecasts of an LDS with "optimal" parameters.

Despite the apparent simplicity of the system (1)-(2), most of the above methods do not apply to this system. This is due to the fact that most of the methods are designed for time invariant, asymptotically stationary systems, where the observation operator ($u_t$ in our notation) is constant and the Kalman gain (or, equivalently $C_t u_t$ in eq. (3)) converges with $t$. In particular this limitation exists in all the system identification results cited above, and is essential to the approaches taken there. However, if the observation vector sequence $u_t$ changes with time – a necessary property for the dynamic regression problem – the system will no longer be asymptotically stationary. In particular, due to this reason, neither the subspace identification methods, nor any of the improper learning approaches above apply to system (1)-(2) .

Among the methods that do apply to (1)-(2) are the general MLE estimation, and some of the auto-correlation methods (Belanger, 1974; Dunik et al., 2018). On one hand, both types of approaches may be applicable to systems apriori more general than (1)-(2). On the other hand, the situation with consistency guarantees – the guarantee that one recovers true parameters given enough observations – for these methods is somewhat complicated. Due to the non-convexity of the likelihood function, the MLE method is not guaranteed to find the true maximum, and as a result the whole method has no guarantees. The results in Belanger (1974); Dunik et al. (2018) do have *asymptotic* consistency guarantees. However, these rely on some explicit and implicit assumptions about the system, the sequence $u_t$ in our case, which can not be easily verified. In particular, Belanger (1974); Dunik et al. (2018) assume *uniform observability* of the system, which we do not assume, and in addition rely on certain implicit assumption about invertibility and condition number of the matrices related to the sequence $u_t$. Moreover, even if one assumes that the assumptions hold, the results are purely asymptotic, and for any finite $T$, do not provide a bound of the expected estimation error as a function of $T$ and $\{u_t\}_{t=1}^{T}$.

In addition, as mentioned earlier, MLE methods by definition must assume that the noises are Gaussian (or belong to some other predetermined parametric family) and autocorrelation based methods also strongly use the Gaussianity assumption. Our approach, on the other hand, requires only sub Gaussian noises with independent coordinates. We note that there are straightforward extensions of our methods to certain cases with dependencies. Indeed, the operator analysis part of this paper does not depend on the distribution of the noises. Therefore, to achieve such an extension, one would only need to correspondingly extend the main probabilistic tool, the Hanson-Wright inequality (Hanson and Wright, 1971; Rudelson et al., 2013, see also Section 4 and Supplementary Material Section E). One such extension, for vectors with the *convex concentration* property, was recently obtained in Adamczak (2015).

## 3 Notation

We refer to Bhatia (1997) and Vershynin (2018) as general references on the notation introduced below, for operators and sub Gaussian variables, respectively.

Let $A : \mathbb{R}^n \to \mathbb{R}^m$ be an operator with a singular value decomposition $A = U \cdot Diag(\lambda_1, \ldots, \lambda_s) \cdot W$, where $s \leq \min\{m, n\}$ and $\lambda_1 \geq \lambda_2 \geq \ldots \geq \lambda_s > 0$. Note that singular values are strictly positive by definition (that is, vectors corresponding to the kernel of $A$ do not participate in the decomposition $A = U \cdot Diag(\lambda_1, \ldots, \lambda_s) \cdot W$). The Hilbert-Schmidt (Frobenius) norm is defined as $\|A\|_{HS} = \sqrt{\sum_{i=1}^s \lambda_i^2}$. The nuclear and the operator norms are given by $\|A\|_{nuc} = \sum_{i=1}^s \lambda_i$ and $\|A\|_{op} = \lambda_1$ respectively.

A centered ($\mathbb{E}X = 0$) scalar random variable $X$ is sub-Gaussian with constant $\kappa$, denoted $X \sim SG(\kappa)$, if for all $t > 0$ it satisfies $\mathbb{P}(|X| > t) \leq 2\exp\left(-\frac{t^2}{\kappa^2}\right)$. A random vector $X = (X_1, \ldots, X_m)$ is $\kappa$ sub-Gaussian, denoted $X \sim SG_m(\kappa)$, if for every $v \in \mathbb{R}^m$ with $|v| = 1$ the random variable $\langle v, X \rangle$ is $\kappa$ sub-Gaussian. A random vector $X$ is $\sigma$-isotropic if for every $v \in \mathbb{R}^m$ with $|v| = 1$, $\mathbb{E}\langle v, X \rangle = \sigma^2$.

Finally, a random vector $X = (X_1, \ldots, X_m)$ is $\sigma$-isotropically $\kappa$ sub-Gaussian with independent components, denoted $X \sim ISG_m(\sigma, \kappa)$ if $X_i$ are independent, and for all $i \leq m$, $\mathbb{E}X_i = 0$, $\mathbb{E}X_i^2 = \sigma^2$ and $X_i \sim SG(\kappa)$. Clearly, if $X \sim ISG_m(\sigma, \kappa)$ then $X$ is $\sigma$-isotropic. Recall also that $X \sim ISG_m(\sigma, \kappa)$ implies $X \sim SG_m(\kappa)$ (Vershynin, 2018). The noise variables we discuss in this paper are $ISG(\kappa, \sigma)$.

Throughout the paper, absolute constants are denoted by $c, c', c'', \ldots$ etc. Their values may change from line to line.

## 4 Overview of the approach

We begin by rewriting (1)-(2) in a vector form. To this end, we first encode sequences of $T$ vectors in $\mathbb{R}^n$, $\{a_t\}_{t \leq T} \subset \mathbb{R}^n$, as a vector $a \in \mathbb{R}^{Tn}$, constructed by concatenation of $a_t$'s. Next, we define the summation operator $S' : \mathbb{R}^T \to \mathbb{R}^T$ which acts on any vector $(h_1, h_2, \ldots, h_T) \in \mathbb{R}^T$ by

$$S'(h_1, h_2, \ldots, h_T) = (h_1, h_1 + h_2, \ldots, \sum_{i \leq T-1} h_i, \sum_{i \leq T} h_i). \tag{6}$$

Note that $S'$ is an invertible operator. Next, we similarly define the summation operator $S : \mathbb{R}^{Tn} \to \mathbb{R}^{Tn}$, an $n$-dimensional extension of $S'$, which sums $n$-dimensional vectors. Formally, for $(h_l)_{l=1}^{Tn} \in \mathbb{R}^{Tn}$, and for $1 \leq j \leq n, 1 \leq t \leq T$, $(Sh)_{(t-1)\cdot n + j} = \sum_{i \leq t} h_{(i-1)\cdot n + j}$. Observe that if the sequence of process noise terms $h_1, \ldots, h_T \in \mathbb{R}^n$ is viewed as a vector $h \in \mathbb{R}^{Tn}$, then by definition $Sh$ is the $\mathbb{R}^{Tn}$ encoding of the sequence $X_t$.

Next, given a sequence of observation vectors $u_1, \ldots, u_T \in \mathbb{R}^n$, we define the observation operator $O_u : \mathbb{R}^{Tn} \to \mathbb{R}^T$ by $(O_u x)_t = \langle u_t, (x_{(t-1)\cdot n + 1}, \ldots, x_{(t-1)\cdot n + n}) \rangle$. In words, coordinate $t$ of $O_u x$ is the inner product between $u_t$ and $t$-th part of the vector $x \in \mathbb{R}^{Tn}$. Define also $Y = (Y_1, \ldots, Y_T) \in \mathbb{R}^T$ to be the concatenation of $Y_1, \ldots, Y_T$. With this notation, one may equivalently rewrite the system (1)-(2) as follows:

$$Y = O_u S h + z, \tag{7}$$

where $h$ and $z$ are independent zero-mean random vectors in $\mathbb{R}^{Tn}$ and $\mathbb{R}^T$ respectively, with independent sub Gaussian coordinates. The variance of each coordinate of $h$ is $\sigma^2$ and each coordinate of $z$ has variance $\eta^2$.

Up to now, we have reformulated our data model as a single vector equation. Note that in that equation, the observations $Y$ and both operators $O_u$ and $S$ are known to us. Our problem may now be reformulated as follows: Given $Y \in \mathbb{R}^T$, assuming $Y$ was generated by (7), provide estimates of $\sigma, \eta$.

As a motivation, we first consider taking the expectation of the norm squared of eq. (7). For any operator $A : \mathbb{R}^m \to \mathbb{R}^m$ and zero-mean vector $h$ with independent coordinates and coordinate variance $\sigma^2$, we have $\mathbb{E}\,|Ah|^2 = \|A\|_{HS}^2\,\sigma^2$, where $\|A\|_{HS}$ is the Hilbert-Schmidt (or Frobenius) norm of $A$. Taking the norm and expectation of (7), and dividing by $T^2$, we thus obtain

$$\frac{\mathbb{E}\,|Y|^2}{T^2} = \frac{\|O_uS\|_{HS}^2}{T^2}\sigma^2 + \frac{T}{T^2}\eta^2. \tag{8}$$

Next, note that $\|O_uS\|_{HS}^2$ is known, and an elementary computation shows that $\frac{\|O_uS\|_{HS}^2}{T^2}$ is of constant order (as a function of $T$; see (25)), while the coefficient of $\eta^2$ is $\frac{1}{T}$. Thus, if the quantity $\frac{|Y|^2}{T^2}$ were close enough to its expectation with high probability, we could take this quantity as a (slightly biased) estimator of $\sigma^2$. However, as it will become apparent later, the deviations of $\frac{|Y|^2}{T^2}$ around the expectation are also of constant order, and thus $\frac{|Y|^2}{T^2}$ can not be used as an estimator. The reason for these high deviations of $\frac{|Y|^2}{T^2}$ is that the spectrum of $O_uS$ is extremely peaked. The highest squared singular value of $O_uS$ is of order $T^2$, the same order as sum of all of them, $\|O_uS\|_{HS}^2$. Contrast this with the case of identity operator, $Id : \mathbb{R}^{Tn} \to \mathbb{R}^{Tn}$: We have $\mathbb{E}\,|Id(h)|^2 = \mathbb{E}\,|h|^2 = Tn\sigma^2$, and one can also easily show that, for instance, $Var\,|Id(h)|^2 = Tn\sigma^2$, and thus the deviations are of order $\sqrt{Tn}\sigma$ – a smaller order than $\mathbb{E}\,|Id(h)|^2$. While for the identity operator the computation is elementary, for a general operator $A$ the situation is significantly more involved, and the bounds on the deviations of $|Y|^2$ will be obtained from the Hanson-Wright inequality (Hanson and Wright, 1971, see also Rudelson et al. (2013)), combined with standard norm deviation bounds for isotropic sub Gaussian vectors.

With these observations in mind, we proceed to flatten the spectrum of $O_uS$ by taking the pseudo-inverse. Let $R : \mathbb{R}^T \to \mathbb{R}^{Tn}$ be the pseudo-inverse, or Moore-Penrose inverse of $O_uS$. Specifically, let

$$O_uS = U \circ Diag(\gamma_1, \ldots, \gamma_T) \circ W, \tag{9}$$

be the singular value decomposition of $O_uS$, where $\gamma_1 \geq \gamma_2 \geq \ldots \geq \gamma_T$ are the singular values.

For the rest of the paper, we will assume that all of the observation vectors $u_t$ are non-zero. This assumption is made solely for notational simplicity and may easily be avoided, as discussed later in this section. Under this assumption, since $S$ is invertible and $O_u$ has rank $T$, we have $\lambda_t > 0$ for all $t \leq T$. For $i \leq T$, denote $\chi_i = \gamma_{T+1-i}^{-1}$. Then $\chi_i$ are the singular values of $R$, arranged in a non-increasing order, and we have by definition

$$R = W^* \circ Diag(\chi_T, \chi_{T-1}, \ldots, \chi_2, \chi_1) \circ U^*, \tag{10}$$

where $W^*, U^*$ denote the transposed matrices of $U, V$, defined in (9).

Similarly to Eq. (8), we apply $R$ to (7), and since $\|RO_uS\|_{HS} = T$, by taking the expectation of the squared norm we obtain

$$\frac{|RY|^2}{T} = \sigma^2 + \frac{\|R\|_{HS}^2}{T}\eta^2 + \left(\frac{|RY|^2}{T} - \frac{\mathbb{E}\,|RY|^2}{T}\right). \tag{11}$$

In this equation, the deviation term $\left(\frac{|RY|^2}{T} - \frac{\mathbb{E}|RY|^2}{T}\right)$ is of order $O(\frac{1}{\sqrt{T}})$ with high probability (Theorem 1). Moreover, the coefficient of $\sigma^2$ is 1, and the coefficient of $\eta^2$, which is $\frac{\|R\|_{HS}^2}{T}$, is of order at least $\Omega(\frac{1}{\log^2 T})$ (Theorem 3, see Section 5 for additional details) – much larger order than $\frac{1}{\sqrt{T}}$. Since $|RY|^2$ and $\|R\|_{HS}^2$ are known, it follows that we have obtained one equation satisfied by $\sigma^2$ and $\eta^2$ up to an error of $\frac{1}{\sqrt{T}}$, where both coefficients are of order larger than the error.

**Algorithm 1** Spectrum Thresholding Variance Estimator (STVE)

1: **Input:** Observations $Y_t$, observation vectors $u_t$, with $t \leq T$, and $p = \alpha T$.
2: Compute the SVD of $O_u S$,

$$O_u S = U \circ Diag(\gamma_1, \ldots, \gamma_T) \circ W,$$

where $\gamma_1 \geq \gamma_2 \geq \ldots \geq \gamma_T > 0$. Denote $\chi_i = \gamma_{T+1-i}^{-1}$ for $1 \leq i \leq T$.
3: Construct the operators

$$R = W^* \circ Diag(\chi_T, \ldots, \chi_1) \circ U^*$$

and

$$R' = W^* \circ Diag(0, 0, \ldots, 0, \chi_p, \ldots, \chi_1) \circ U^*$$

4: Produce the estimates:

$$\widehat{\eta^2} = \left( \frac{|R'Y|^2}{p} - \frac{|RY|^2}{T} \right) \bigg/ \left( \frac{\|R'\|_{HS}^2}{p} - \frac{\|R\|_{HS}^2}{T} \right)$$

$$\widehat{\sigma^2} = \frac{|RY|^2}{T} - \frac{\|R\|_{HS}^2}{T} \widehat{\eta^2}.$$

Next, we would like to obtain another linear relation between $\sigma^2$, $\eta^2$. To this end, choose some $p = \alpha T$, where $0 < \alpha < 1$ is of constant order. The possible choices of $p$ are discussed later in this section. We define an operator $R' : \mathbb{R}^T \to \mathbb{R}^{Tn}$ to be a version of $R$ truncated to the first $p$ singular values. If (10) is the SVD decomposition of $R$, then

$$R' = W^* \circ Diag(0, 0, \ldots, \chi_p, \chi_{p-1}, \ldots, \chi_1) \circ U^*.$$

Similarly to the case for $R$, we have

$$\frac{|R'Y|^2}{p} = \sigma^2 + \frac{\|R'\|_{HS}^2}{p} \eta^2 + \left( \frac{|R'Y|^2}{p} - \frac{\mathbb{E}|R'Y|^2}{p} \right). \tag{12}$$

The deviations in (12) are also described by Theorem 1. Note also that since $\|R'\|_{HS}^2$ is the sum of $p$ largest squared singular values of $R$, by definition it follows that $\frac{\|R'\|_{HS}^2}{p} \geq \frac{\|R\|_{HS}^2}{T}$.

Now, given two equations in two unknowns, we can solve the system to obtain the estimates $\widehat{\sigma^2}$ and $\widehat{\eta^2}$. The full procedure is summarized in Algorithm 1, and the bounds implied by Theorem 1 on the estimators $\widehat{\sigma^2}$ and $\widehat{\eta^2}$ are given in Corollary 2. We first state these results, and then discuss in detail the various parameters appearing in the bounds.

**Theorem 1.** *Consider a random vector $Y \in \mathbb{R}^T$ of the form $Y = O_u Sh + z$ where $h \sim ISG_{Tn}(\sigma, \kappa)$ and $z \sim ISG_T(\eta, \kappa)$. Set $|u_{min}| = \min_t |u_t|$. Then for any $0 < \delta < 1$,*

$$\mathbb{P}\left( \left| \frac{|RY|^2}{T} - \left( \sigma^2 + \frac{\|R\|_{HS}^2}{T} \eta^2 \right) \right| \geq c \frac{B}{\sqrt{T}} \right) \leq 4\delta, \tag{13}$$

$$\mathbb{P}\left( \left| \frac{|R'Y|^2}{p} - \left( \sigma^2 + \frac{\|R'\|_{HS}^2}{p} \eta^2 \right) \right| \geq c \frac{B}{\sqrt{p}} \right) \leq 4\delta \tag{14}$$

*where $B$ is given by*

$$B = \left(1 + \kappa^2\right) \left(1 + |u_{min}|^{-2}\right) \log \frac{1}{\delta}. \tag{15}$$

The bounds on the estimators of Algorithm 1 are given in the following Corollary. As discussed below, in addition to Theorem 1, the key to the derivation of this Corollary are the estimates of the spectrum of $R$, given in Theorem 3, Section 5.

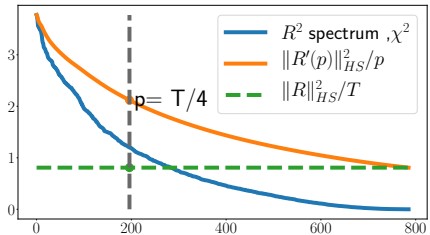
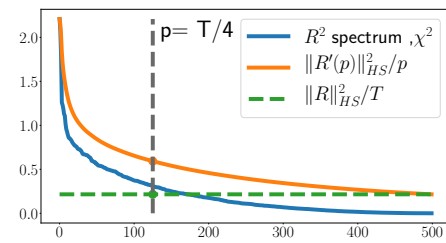

(a) Spectrum ($\chi_i^2$) of $R$ for electricity data (blue). $\|R'(p)\|_{HS}^2 / p$ as a function of $p$ (orange). The mean $\|R\|_{HS}^2 / T$ (green).

(b) Similar figure for synthetic data, $T = 500$.

Figure 1: Spectra of $R$

**Corollary 2.** *Let* $\widehat{\sigma^2}, \widehat{\eta^2}$ *be the estimators of* $\sigma^2, \eta^2$ *obtained form Algorithm 1 with* $p \geq \frac{1}{4}T$. *Set* $|u_{max}| = \max_t |u_t|$. *Then for any* $0 < \delta < 1$, *with probability at least* $1 - 8\delta$,

$$\left|\widehat{\sigma^2} - \sigma^2\right| \leq c\frac{B}{\sqrt{T}} \left(1 - \frac{p\|R\|_{HS}^2}{T\|R'\|_{HS}^2}\right)^{-1}, \tag{16}$$

$$\left|\widehat{\eta^2} - \eta^2\right| \leq c\frac{B}{\sqrt{T}} \left(1 - \frac{p\|R\|_{HS}^2}{T\|R'\|_{HS}^2}\right)^{-1} n^2 |u_{max}|^2 \log^2 T, \tag{17}$$

*with B given by* (15).

We first discuss the assumption $|u_{min}| > 0$. This assumption is made solely for notational convenience, as detailed below. To begin, note that some form of lower bound on the norms of the observation vectors $u_t$ *must* appear in the bounds. This is simply because if one had $u_t = 0$ for all $T$, then clearly no estimate of $\sigma$ would have been possible. On the other hand, our use of the smallest value $|u_{\min}|$ may seem restrictive at first. We note however, that instead of considering the observation operator $O_u : \mathbb{R}^{Tn} \to \mathbb{R}^T$, one may consider the operator $O_{\bar{u}} : \mathbb{R}^{Tn} \to \mathbb{R}^{\bar{T}}$ for any subsequence $\{\bar{u}_{\bar{t}}\}_{\bar{t}=1}^{\bar{T}}$. The observation vector $Y$ would be correspondingly restricted to the subsequence of indices. This allows us to treat missing values and to exclude any outlier $u_t$ with small norms. All the arguments in Theorems 1 and 3 hold for this modified $O_{\bar{u}}$ without change. The only price that will be paid is that $T$ will be replaced by $\bar{T}$ in the bounds. Moreover, we note that typically we have $|u_t| \geq 1$ by construction, see for instance Section 6.2. Additional discussion of missing values may be found in Supplementary Material Section H.

Next, up to this point, we have obtained two equations, (11)-(12), in two unknowns, $\sigma^2, \eta^2$. Note that in order to be able to obtain $\eta^2$ from these equations, at least one of the coefficients of $\eta^2$, either $\frac{\|R\|_{HS}^2}{T}$ or $\frac{\|R'\|_{HS}^2}{p}$ must be of larger order than $\frac{1}{\sqrt{T}}$, the order of deviations. Providing lower bounds on these quantities is one of the main technical contributions of this work. This analysis uses the connection between the operator $S$ and the Laplacian on the line, and resolves the issue of translating spectrum estimates for the Laplacian into the spectral estimates for $R$. We note that there are no standard tools to study the spectrum of $R$, and our approach proceeds indirectly via the analysis of the nuclear norm of $O_u S$. These results are stated in Theorem 3. In particular, we show that $\frac{\|R\|_{HS}^2}{T}$ is $\Omega(\frac{1}{\log^2 T})$, which is the source of the log factor in (17).

Finally, in order to solve the equations (11)-(12), not only the equations must have large enough coefficients, but the equations must be *different*. This is reflected by the term $\left(1 - \frac{p\|R\|_{HS}^2}{T\|R'\|_{HS}^2}\right)^{-1}$ in (16), (17). Equivalently, while $\frac{\|R'\|_{HS}^2/p}{\|R\|_{HS}^2/T} \geq 1$ by definition, we would like to have

$$\frac{\|R'\|_{HS}^2 / p}{\|R\|_{HS}^2 / T} \geq 1 + const \tag{18}$$

for the bounds (16), (17) to be stable. Note that since both $\|R'\|_{HS}^2$ and $\|R\|_{HS}^2$ are computed in Algorithm 1, the condition (18) can simply be verified before the estimators $\widehat{\sigma^2}, \widehat{\eta^2}$ are returned.

It is worth emphasizing that for simple choices of $p$, say $p = \frac{1}{4}T$, the condition (18) does hold in practice. Note that, for any $p$, we can have $\frac{\|R'\|_{HS}^2/p}{\|R\|_{HS}^2/T} = 1$ only if the spectrum of $R$ is *constant*. Thus (18) amounts to stating that the spectrum of $R$ exhibits some decay. As we show in experiments below, the spectrum (squared) of $R$, for $u_t$ derived from daily temperature features, or for random Gaussian $u_t$, indeed decays. See Section 6, Figures 1a and 1b. In particular, in both cases (18) holds with $const > 1$. Additional bounds on the quantity $\left(1 - \frac{p\|R\|_{HS}^2}{T\|R'\|_{HS}^2}\right)^{-1}$ under various assumptions on the sequence $u_t$ are given in Section G of the Supplementary Material.

## 5 Properties of $O_uS$ and $R$

As discussed in Section 4 (see the discussion following eq. (11)), one of the crucial points enabling Algorithm 1 and its analysis is the fact that the quantity $\frac{\|R\|_{HS}^2}{T}$ is bounded below by an expression that is of much higher order than the noise magnitude $\frac{1}{\sqrt{T}}$.

In this section we provide the formal statement of this and other associated results, and discuss the related arguments. First, we obtain the following bound on the spectrum of $O_uS$ (Lemma 4, Supplementary Material Section B). Recall that the nuclear norm was defined in Section 3, and that for a sequence $u_t$ we set $|u_{max}| = \max_t |u_t|$ and $|u_{min}| = \min_t |u_t|$. Then:

$$\|O_uS\|_{nuc} = \sum_{t \leq T} \lambda_t(O_uS) \leq 4n\,|u_{max}|\,T \log T. \tag{19}$$

The proof of this bound exploits the connection between $S$ and the Laplacian on the line. In particular, we use the fact that the eigenvalues of the Laplacian are known precisely, satisfying $\lambda_l = 2\sin\left(\frac{\pi(T-l)}{2T}\right)$. Next, we state the lower (and upper) bounds for $R$.

**Theorem 3.** *Let* $R : \mathbb{R}^T \to \mathbb{R}^{Tn}$ *be the pseudoinverse of* $O_uS$. *Then*

$$c\frac{1}{n\,|u_{max}|\log T} \leq \|R\|_{op} \leq 2\,|u_{min}|^{-1}, \tag{20}$$

$$c\frac{1}{n^2\,|u_{max}|^2 \log^2 T}T \leq \|R\|_{HS}^2 \leq 4\,|u_{min}|^{-2}\,T, \tag{21}$$

$$c\frac{1}{n^4\,|u_{max}|^4 \log^4 T}T \leq \|R^*R\|_{HS}^2 \leq 16\,|u_{min}|^{-4}\,T. \tag{22}$$

Due to the complicated structure of $R$ as a pseudo-inverse of a composition of operators, there are no direct ways to control individual eigenvalues of $R$. Thus the main technical issue resolved in Theorem 3 is nevertheless obtaining lower bounds on $\|R\|_{HS}^2$. Our approach is rather indirect, and we obtain these bounds from the nuclear norm bound (19) via a Markov type inequality on the eigenvalues.

## 6 Experiments

### 6.1 Synthetic Data

In this section the performance of STVE is evaluated on synthetic data. The data was generated by the LDS (1)-(2), using Gaussian noises with $\sigma^2 = 0.5, \eta^2 = 2$. The input dimension was $n = 5$, and the input sequence $u_t$ sampled from the Gaussian $N(0, I_n)$.

We run the STVE algorithm for different values of $T$, where for each $T$ we sampled the data 150 times. Figure 2a shows the average (over 150 runs) estimation error for both process and observation noise variances for various values of $T$. As expected from the bounds in Corollary 2, it may be observed in Figure 2a that the estimation errors decay roughly at the rate of $const/\sqrt{T}$. A typical spectrum of $R$ is shown in Figure 1b. For larger $T$, the spectra also exhibits similar decay.

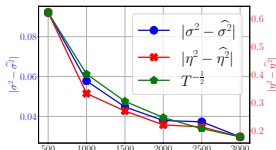 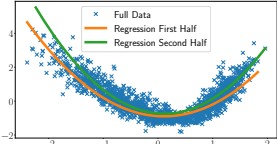 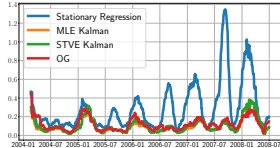

(a) STVE variance estimation errors vs $T$, and the $T^{-\frac{1}{2}}$ decay.

(b) Load (y-axis) against Temperature (x-axis), both axis normalized. The full data (blue points), regression learned on the first half (orange), regression learned on the second half (green).

(c) Smoothed prediction errors. Stationary Regression trained on first half (blue), MLE (orange), STVE (green), OG (red).

Figure 2: Evaluation

## 6.2 Temperatures and Electricity Consumption

In this section we examine the relation between daily temperatures and electricity consumption in the data from Hong et al. (2014) (see also Hong (2016)). The following forecasting methods are compared: a stationary regression, an online gradient, and a Kalman filter for a dynamic regression, with parameters learned via MLE or STVE. We find that the Kalman filter methods provide the best performance, with no significant difference between STVE and MLE derived systems.

The data consists of total daily electricity consumption (load) $y_t$, and the average daily temperature, $v_t$, for a certain region, for the period Jan-2004 to Jun-2008. Full details on the preprocessing of the data, as well as additional details on the experiments, are given in Supplementary Material Section I. Here we note that the data contains missing load values, for 9 non-consecutive weeks (out of about 234 weeks total). All methods discussed here, including STVE, can naturally incorporate missing values, as discussed in Supplementary Material Section H.

An elementary inspection of the data reveals that the load may be reasonably approximated as a quadratic function of the temperature, $y_t = x_{t,1} \cdot 1 + x_{t,2} \cdot v_t + x_{t,3} \cdot v_t^2$, where $u_t = (1, v_t, v_t^2)$ is the observation vector (features), and $x_t = (x_{t,1}, x_{t,2}, x_{t,3})$ is the possibly time varying regression vector. This is shown in Figure 2b, where we fit a stationary (time invariant) regression of the above form, using either only the first or only the second half of the data. We note that these regressions *differ* – the regression vector changes with time. It is therefore of interest to track it via online regression.

We use the first half of the data (train set) to learn the parameters $\sigma, \eta$ of the online regression (1)-(2) via MLE optimization and using STVE. We also use the train set to find the optimal learning rate $\alpha$ for the OG forecaster described by the update equation (5). This learning rate is chosen as the rate that yields smallest least squares forecast error on the train set. In addition, we learn a time independent, stationary regression on the first half of the data.

We then employ the learned parameters to make predictions of the load given the temperature, by all four methods. The predictions for the system (1)-(2) are made with a Kalman filter (at time $t$, we use the filtered state estimate $\tilde{x}_t$, which depends only on $y_1, \ldots, y_t$ and $u_1, \ldots, u_t$, and make the prediction $\tilde{y}_{t+1} = \langle \tilde{x}_t, u_{t+1} \rangle$).

Daily squared prediction errors (that is, $(y_t - \tilde{y}_t)^2$) are shown in Figure 2c (smoothed with a moving average of 50 days). We see that the adaptive models (MLE, STVE, OG) outperform the stationary regression already on the train set (first half of the data), and that the difference in performance becomes dramatic on the second half (test). It is also interesting to note that the performance of the Kalman filter based methods (MLE, STVE) is practically identical, but both are somewhat better than the simpler OG approach.

We also note that by construction, we have $|u_{min}| \geq 1$ in this experiment, due to the constant 1 coordinate, and also $|u_{max}| \leq 5$, due to the normalization. Since these operations are typical for any regression problem, we conclude that the direct influence of $|u_{min}|$ and $|u_{max}|$ on the bounds in Theorem 1 and Corollary 2 will not usually be significant.

# 7 Conclusion And Future Work

In this work we introduced the STVE algorithm for estimating the variance parameters of LDSs of type (1)-(2), and obtained the first sample complexity guarantees for such estimators. We have also shown how the shape of the spectrum of $R$ can be exploited to obtain the estimators and the related bounds, thus providing the first explicit geometric parameter of the data that affects the bounds.

As discussed in Section 1 and demonstrated in Section 6, the system (1)-(2) is of independent interest in applications. However, we also believe that the analysis presented here is an important first step towards a finite time data-dependent quantitative understanding of general LDSs. and perhaps even non-linear dynamical systems.

## Acknowledgments and Disclosure of Funding

This research was supported by the ISRAEL SCIENCE FOUNDATION (grant No. 2199/20).

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
