Figure 3: Lag and Overfitting to the most recent observation, for various variance values. See the discussion in Sections 1 and A.

# Finite Sample Analysis Of Dynamic Regression Parameter Learning - Supplementary Material

## A   Outline

This Supplementary Material is organized as follows: The proofs of the results of Section 5, including Theorem 3, as well as proofs of Theorems 1, and Corollary 2, are given in Sections C to F. In Section G we state and prove additional bounds on the quantity $\left(1 - \frac{p\|R\|_{HS}^2}{T\|R'\|_{HS}^2}\right)^{-1}$ that appears in Corollary 2. Section H contains the discussion of missing values in STVE. Additional details on the electricity consumption experiment are given in Section I.

Finally, we describe Figure 3. The data (observations, red) was generated from the system (1)-(2) with one dimensional state ($n = 1$), and we set $u_t = 1$. The states produced by the Kalman filter with ground truth values of $\sigma, \eta$ are shown in green, while states obtained from other choices of parameters are shown in black and blue.

## B   Bounds on $O_u S$, Lemma 4

In this Section we prove the following bounds on the spectrum of $O_u S$. See Section 5 for a discussion of these bounds.

**Lemma 4.** *The singular values of $O_u S$ satisfy the following:*

$$\lambda_1(O_u S) \le |u_{max}| \cdot T \text{ and } \lambda_T(O_u S) \ge \frac{1}{2} |u_{min}| \tag{23}$$

$$\|O_u S\|_{nuc} = \sum_{t \le T} \lambda_t(O_u S) \le 4n |u_{max}| T \log T \tag{24}$$

$$\frac{1}{4} |u_{min}|^2 T^2 \le \|O_u S\|_{HS}^2 = \sum_{t \le T} t |u_t|^2 \le \frac{1}{2} |u_{max}|^2 T^2. \tag{25}$$

The proof of this lemma uses the following auxiliary result: Let $D_T : \mathbb{R}^T \to \mathbb{R}^{T-1}$ be the difference operator, $(D_T x)_t = x_{t+1} - x_t$ for $t \le T - 1$. In the field of Finite Difference methods, the operator $D_T D_T^*$ is known as the Laplacian on the line, or as the discrete derivative with Dirichlet boundary conditions, and is well studied. The eigenvalues of $D_T D_T^*$ may be derived by a direct computation, and correspond to the roots of the Chebyshev polynomial of second kind of order $T$. In particular, the following holds:

**Lemma 5.** *The operator $D_T$ has kernel of dimension $1$ and singular values $\lambda_l(D_T) = 2\sin\left(\frac{\pi(T-l)}{2T}\right)$ for $l = 1, \dots, T-1$.*

We refer to Mitchell and Griffiths (1980) for the proof of Lemma 5. Next, in Lemma 6, we show that the inverse of the operator $S'$, defined in (6), is a one dimensional perturbation of $D_T$, which implies bounds on singular values of $S'^{-1}$.

**Lemma 6.** *The singular values of $S_T'^{-1}$ satisfy*

$$2\sin\left(\frac{\pi(T-t)}{2(T+1)}\right) \le \lambda_t(S_T'^{-1}) \le 2\sin\left(\frac{\pi(T+1-t)}{2(T+1)}\right) \tag{26}$$

*for $1 \le t \le T-1$, and*

$$\frac{1}{T} \le \lambda_T(S_T'^{-1}) \le 2\sin\left(\frac{\pi}{2(T+1)}\right). \tag{27}$$

The proof of this is given in the next section. With the estimates of Lemma 6, the bounds in Lemma 4 follow. Proof of Lemma 4:

*Proof.* By Lemma 6,

$$\|S\|_{op} \le T \text{ and } |S_T'x| \ge \frac{1}{2}|x| \text{ for all } x \in \mathbb{R}^T. \tag{28}$$

Note also that $S$ is by definition a collection of $n$ independent copies of $S_T'$, and therefore the spectrum of $S$ is that of $S_T'$, but each singular value is taken with multiplicity $n$. In particular it follows that (28) holds also for $S$ itself. Since clearly $\|O_u\|_{op} \le |u_{max}|$, the upper bound on $\|O_u S\|_{op}$ in (23) follows from (28).

For the lower bound, denote by $V'$ the orthogonal complement to the kernel of $O_u$, $V' = (Ker(O_u))^\perp$. Denote by $P_{V'} : \mathbb{R}^{Tn} \to V'$ the orthogonal projection onto $V'$. We have in particular that $O_u S = O_u P_{V'} S$. Next, the operator $S$ maps the unit ball $B_{Tn}$ of $\mathbb{R}^{Tn}$ into an ellipsoid $\mathcal{E}_T$, and by (28), we have $\frac{1}{2} B_{Tn} \subset \mathcal{E}_T$. It therefore follows that

$$\frac{1}{2} B_{V_1} \subset (P_{V'} S)(B_{Tn}), \tag{29}$$

where $B_{V_1}$ is the unit ball of $V_1$. It remains to observe that for every $x \in V_1$, we have

$$|O_u x| \ge |u_{min}| |x|. \tag{30}$$

Combining (29) and (30), we obtain the lower bound in (23).

To derive (24), recall that the nuclear norm is sub-multiplicative with respect to the operator norm:

$$\|O_u S\|_{nuc} \le \|O_u\|_{op} \cdot \|S\|_{nuc} \le |u_{max}| \cdot \|S\|_{nuc}. \tag{31}$$

This follows for instance from the characterization of the nuclear norm as trace dual of the operator norm (Bhatia, 1997, Propositions IV.2.11, IV.2.12). Next, since the spectrum of $S$ is the spectrum of $S'$ taken with multiplicity $n$, we have

$$\|S\|_{nuc} = n \|S'\|_{nuc}, \tag{32}$$

and it remains to bound the nuclear norm of $S'$.

Using the inequality

$$\sin\frac{\pi}{2}\alpha \ge \alpha \text{ for all } \alpha \in [0,1], \tag{33}$$

and Lemma 6, we have

$$\|S'\|_{nuc} = T + 2\sum_{t \le T-1} \sin^{-1}\left(\frac{\pi T - t}{2(T+1)}\right) \le 2T \sum_{t \le T} \frac{1}{t+1} \le 4T \log T. \tag{34}$$

Combining (31), (32) and (34), the inequality (24) follows.

It remains to estimate the Hilbert-Schmidt norm of $O_u S$, which can be done by a direct computation. Recall that for any operator $A : \mathbb{R}^m \to \mathbb{R}^m$ the Hilbert-Schmidt norm satisfies

$$\|A\|_{HS}^2 = tr A^* A = \sum_{i \le m} \langle A^* A\phi_i, \phi_i \rangle = \sum_{i \le m} \langle A\phi_i, A\phi_i \rangle = \sum_{i \le m} |A\phi_i|^2, \tag{35}$$

for any orthonormal basis $\phi_i$ in $\mathbb{R}^m$. Let $e_{ti}$ be the standard basis vector in $\mathbb{R}^{Tn}$ corresponding to coordinate $i \le n$ at time $t \le T$. Let $e_{t'}$, $t' \le T$ denote the standard basis in $\mathbb{R}^T$. Then

$$O_u S e_{ti} = \sum_{t'=t}^{T} u_{t'i} e_{t'}, \tag{36}$$

where $u_{t'i}$ is the $i$-th coordinate of $u_{t'}$. It follows that

$$|O_u S e_{ti}|^2 = \sum_{t'=t}^{T} u_{t'i}^2 \tag{37}$$

and hence

$$\|O_u S\|_{HS}^2 = \sum_{t \le T, i \le n} |O_u S e_{ti}|^2 = \sum_{t \le T} \sum_{t'=t}^{T} |u_{t'}|^2 = \sum_{t \le T} t \, |u_t|^2. \tag{38}$$

The bounds (25) follow directly from (38). $\qquad\square$

## C  Proof of Lemma 6

*Proof.* Recall that the operator $S_T'^{-1}$ is given by $S_T'^{-1} x = (x_1, x_2 - x_1, \ldots, x_T - x_{T-1})$ and the operator $D = D_T : \mathbb{R}^T \to \mathbb{R}^{T-1}$ is given by $Dx = (x_2 - x_1, \ldots, x_T - x_{T-1})$. Let $V = \mathrm{span}\{e_2, \ldots, e_T\}$ be the subspace spanned by all but the first coordinate. Let $P_V : \mathbb{R}^T \to \mathbb{R}^T$ be the projection onto $V$, i.e. a restriction to second to $T$'th coordinate. Observe that the action of $DP_V$ is equivalent to that of $S_{T-1}'^{-1}$. Therefore the singular values of $S_{T-1}'^{-1}$ are identical to those of $DP_V$. To obtain bounds on the singular values of $DP_V$, note that $(DP_V)^* DP_V = P_V D^* DP_V$ – that is, $P_V D^* DP_V$ is a *compression* of $D^*D$. Thus, by the Cauchy's Interlacing Theorem (Bhatia, 1997, Corollary III.1.5),

$$\lambda_t(D^*D) \ge \lambda_t(P_V D^* DP_V) \ge \lambda_{t+1}(D^*D) \tag{39}$$

for all $t \le T-1$. In conjunction with Lemma 5 this provides us with the estimates for all but the smallest singular value of $S_{T-1}'^{-1}$ (since $\lambda_T(D^*D) = 0$). We therefore estimate $\lambda_{T-1}(S_{T-1}'^{-1}) = \lambda_T(DP_V)$ directly, by bounding the norm of $S_{T-1}'$. Indeed, for any $T$, by the Cauchy-Schwartz inequality,

$$|S_T' x|^2 = \sum_{t \le T} \left( \sum_{i \le t} x_i \right)^2 \le \sum_{t \le T} \left( \sum_{i \le t} x_i^2 \right) \left( \sum_{i \le t} 1 \right) \tag{40}$$

$$= \sum_{t \le T} t \left( \sum_{i \le t} x_i^2 \right) \le \left( \sum_{t \le T} t \right) |x|^2 \le |x|^2 \, T^2. \tag{41}$$

Thus we have $\|S_T'\|_{op} \le T$, which concludes the proof of the Lemma. $\qquad\square$

## D  Proof of Theorem 3

*Proof.* The bound on $\|R\|_{op}$ follows directly from the lower bound on the singular values of $O_u S$ in (23). Since $R$ is of rank $T$, the upper bounds on $\|R\|_{HS}$ follow directly from the $\|R\|_{op}$ bound.

The lower bounds on $\|R\|_{HS}^2$ and $\|R^* R\|_{HS}^2$ follow from the upper bounds on the nuclear norm in Lemma 4. Note that this argument would not have worked if we only had upper bounds on the Hilbert-Schmidt norm, rather than the nuclear norm in Lemma 4. Denote $\gamma_i = \lambda_i(O_u S)$. From (24) in Lemma 4, the number of $\gamma_i$ that are larger than $4n \, |u_{max}| \log T$ satisfies

$$\# \{\gamma_i \mid \gamma_i \ge 4n \, |u_{max}| \log T\} \le \frac{T}{2}. \tag{42}$$

Since there are total of $T$ singular values $\gamma_i$ overall, we can equivalently rewrite (42) as

$$\# \left\{ \gamma_i \,\middle|\, \gamma_i^{-1} \ge \frac{1}{4n \, |u_{max}| \log T} \right\} \ge \frac{T}{2}. \tag{43}$$

This immediately implies the lower bounds on $\|R\|_{HS}^2$ and $\|R^* R\|_{HS}^2$. $\qquad\square$

# E Proof of Theorem 1

The two main probabilistic tools that we use are the Hanson-Wright inequality (Hanson and Wright, 1971), and a classical norm deviation inequality for sub Gaussian vectors, as follows:

**Theorem 7** (Hanson-Wright Inequality). *Let $X = (X_1, \ldots, X_m) \in \mathbb{R}^m$ be a random vector such that the components $X_i$ are independent and $X_i \sim SG(\kappa)$ for all $i \leq m$. Let $A$ be an $m \times m$ matrix. Then, for every $t \geq 0$,*

$$\mathbb{P}\left(|\langle AX, X\rangle - \mathbb{E}\langle AX, X\rangle| > t\right) \leq 2\exp\left\{-c\min\left(\frac{t^2}{\kappa^4 \|A\|_{HS}^2}, \frac{t}{\kappa^2 \|A\|_{op}}\right)\right\}. \tag{44}$$

In particular, recall that we are interested in concentration of $|RY|^2$. We may write:

$$|RY|^2 = |RO_u Sh + Rz|^2 = |RO_u Sh|^2 + |Rz|^2 + \langle h, (RO_u S)^* Rz\rangle. \tag{45}$$

The deviations of the first two terms may be bounded via Theorem 7. For the third term, we use the following:

**Lemma 8.** *For any $X \sim SG_m(\kappa)$ we have:*

$$\mathbb{P}\left(|X| > 4\kappa\sqrt{m} + t\right) \leq \exp\left(-\frac{ct^2}{\kappa^2}\right). \tag{46}$$

Note that in Lemma 8 we do not require the coordinates of $X$ to be independent. This will be important in what follows. Lemma 8 is standard and can be proved via covering number estimates of the Euclidean ball, see for instance Vershynin (2018), Section 4.4.

In addition, the following observation is used throughout the text:

**Lemma 9.** *Let $A : \mathbb{R}^n \to \mathbb{R}^m$ be an operator, and let $h = (h_1, \ldots, h_m)$ have independent coordinates with $\mathbb{E}h_i^2 = \sigma^2$. Denote by $\{\lambda_i\}_{i=1}^k$, $k \leq \min(m, n)$, the singular values of $A$. Then*

$$\mathbb{E}|Ah|^2 = \sigma^2 \|A\|_{HS}^2 = \sigma^2 \sum_{i=1}^k \lambda_i^2. \tag{47}$$

The elementary proof is omitted.

We now prove Theorem 1.

*Proof.* Let $0 < \delta < 1$ be given. We first bound the deviations from the expectation for the first term in (45), $|RO_u Sh|^2$. We apply the Hanson-Wright inequality with $X = h$ and $A = (RO_u S)^* RO_u S$. By definition, $A$ has a single eigenvalue 1 with multiplicity $T$. Thus clearly $\|A\|_{HS}^2 = T$ and $\|A\|_{op} = 1$.

For an appropriate constant $c' > 0$, set $t = c'\kappa^2\sqrt{T}\log\frac{1}{\delta}$. Then,

$$\mathbb{P}\left(\left||RO_u Sh|^2 - T\sigma^2\right| \geq c''\kappa^2\sqrt{T\log\frac{1}{\delta}}\right) \leq \delta. \tag{48}$$

The deviation of the second term in (45) is similarly bounded using $A = R^* R$. Recall that by Theorem 3 we have $\|R\|_{op} \leq 2|u_{min}|^{-1}$, and note that $\|R^* R\|_{HS}^2 \leq T\|R\|_{op}^4 \leq cT|u_{min}|^{-4}$. Set $t = c'\kappa^2|u_{min}|^{-2}\sqrt{T}\log\frac{1}{\delta}$. With this choice it follows that both terms in the minimum in (44) are larger than $c\log\frac{1}{\delta}$ and we have

$$\mathbb{P}\left(\left||Rz|^2 - \eta^2\|R\|_{HS}^2\right| \geq c'\kappa^2|u_{min}|^{-2}\sqrt{T}\log\frac{1}{\delta}\right) \leq \delta. \tag{49}$$

Finally, we bound the third term in (45). Denote by $D$ the event

$$D = \left\{|(RO_u S)^* Rz| > c|u_{min}|^{-1}\left(\kappa\sqrt{T} + \kappa\sqrt{\log\frac{1}{\delta}}\right)\right\}, \tag{50}$$

and by $E$ the event

$$E = \left\{ |\langle h, (RO_u S)^* R z \rangle| > c' |u_{min}|^{-1} \left( \kappa \sqrt{T} + \kappa \sqrt{\log \frac{1}{\delta}} \right) \kappa \sqrt{\log \frac{1}{\delta}} \right\}. \tag{51}$$

By Lemma 8 applied to $z$, and using the fact that $\|(RO_u S)^* R\|_{op} \leq 2 |u_{min}|^{-1}$,

$$\mathbb{P}(D) \leq \mathbb{P}\left( |z| > c\kappa\sqrt{T} + c\kappa\sqrt{\log\frac{1}{\delta}} \right) \leq \delta. \tag{52}$$

Next, using independence of $h, z$ and $h \sim SG_{Tn}(\kappa)$,

$$\mathbb{P}(E \mid D^c) \leq \delta \tag{53}$$

where $D^c$ is the complement of $D$. Therefore, combining (52) and (53),

$$\mathbb{P}(E) = \mathbb{P}(D)\,\mathbb{P}(E|D) + \mathbb{P}(D^c)\,\mathbb{P}(E|D^c) \leq 2\delta. \tag{54}$$

Combining (48), (49) and (54), we obtain via the union bound:

$$\mathbb{P}\left( \left| \frac{|RY|^2}{T} - \left( \sigma^2 + \frac{\|R\|_{HS}^2}{T}\eta^2 \right) \right| \geq c\frac{\left(1 + \kappa^2\right)\left(1 + |u_{min}|^{-2}\right)\log\frac{1}{\delta}}{\sqrt{T}} \right) \leq 4\delta. \tag{55}$$

Similarly, we obtain a bound for the equations involving $R'$:

$$\mathbb{P}\left( \left| \frac{|R'Y|^2}{p} - \left( \sigma^2 + \frac{\|R'\|_{HS}^2}{p}\eta^2 \right) \right| \geq c\frac{\left(1 + \kappa^2\right)\left(1 + |u_{min}|^{-2}\right)\log\frac{1}{\delta}}{\sqrt{p}} \right) \leq 4\delta. \tag{56}$$

The only difference in the derivation of (56) compared to (55) is in the application of Lemma 8. In the later case, to replace $\sqrt{T}$ with $\sqrt{p}$ in (52), we apply Lemma 8 with $X = P_V z$ rather than with $X = z$, where $P_V$ is the projection onto the range of $R'$ – a $p$-dimensional space. Note that $P_V z$ does not necessarily have a structure of $p$ independent coordinates, but is sub Gaussian and isotropic. Therefore Lemma 8 still applies.

$\square$

# F   Proof of Corollary 2

We now turn to prove Corollary 2.

*Proof.* Denote

$$E(a) = \frac{\left(1 + \kappa^2\right)\left(1 + |u_{min}|^{-2}\right)\log\frac{1}{\delta}}{a}. \tag{57}$$

Using (56) and (55) we may write

$$\frac{|RY|^2}{T} + e_1 = \sigma^2 + \frac{\|R\|_{HS}^2}{T}\eta^2, \tag{58}$$

$$\frac{|R'Y|^2}{p} + e_2 = \sigma^2 + \frac{\|R'\|_{HS}^2}{p}\eta^2, \tag{59}$$

where $e_1, e_2$ are error terms such that $|e_1| \leq E(\sqrt{T})$ and $|e_2| \leq E(\sqrt{p})$ holds with probability at least $1 - 8\delta$.

It follows that

$$\eta^2 = \left( \frac{|R'Y|^2}{p} - \frac{|RY|^2}{T} \right) \left( \frac{\|R'\|_{HS}^2}{p} - \frac{\|R\|_{HS}^2}{T} \right)^{-1} + (e_2 - e_1)\left( \frac{\|R'\|_{HS}^2}{p} - \frac{\|R\|_{HS}^2}{T} \right)^{-1} \tag{60}$$

and

$$\frac{p\,\|R\|_{HS}^2}{T\,\|R'\|_{HS}^2}\frac{|R'Y|^2}{p} + \frac{p\,\|R\|_{HS}^2}{T\,\|R'\|_{HS}^2}e_2 = \frac{p\,\|R\|_{HS}^2}{T\,\|R'\|_{HS}^2}\sigma^2 + \frac{\|R\|_{HS}^2}{T}\eta^2. \tag{61}$$

Thus

$$\sigma^2 = \left(1 - \frac{p\,\|R\|_{HS}^2}{T\,\|R'\|_{HS}^2}\right)^{-1}\left(\frac{|RY|^2}{T} - \frac{|R'Y|^2}{T}\frac{\|R\|_{HS}^2}{\|R'\|_{HS}^2}\right)$$

$$+ \left(1 - \frac{p\,\|R\|_{HS}^2}{T\,\|R'\|_{HS}^2}\right)^{-1}\left(e_1 - \frac{p\,\|R\|_{HS}^2}{T\,\|R'\|_{HS}^2}e_2\right). \tag{62}$$

It remains to observe that

$$\left(\frac{\|R'\|_{HS}^2}{p} - \frac{\|R\|_{HS}^2}{T}\right)^{-1} = \left(1 - \frac{p\,\|R\|_{HS}^2}{T\,\|R'\|_{HS}^2}\right)^{-1}\frac{p}{\|R'\|_{HS}^2} \tag{63}$$

$$\leq c\left(1 - \frac{p\,\|R\|_{HS}^2}{T\,\|R'\|_{HS}^2}\right)^{-1}n^2\,|u_{max}|^2\log^2 T, \tag{64}$$

where the inequality follows from eq. (43) in the proof of Theorem 3. $\qquad\square$

## G   Thresholding Gap Analysis

The main result of this section is the following Proposition.

**Proposition 10.** *Given the sequence $\{u_t\}_{t=1}^T \subset \mathbb{R}^n$, define the scalar sequence $\tilde{u}_t = \sum_{i\leq n} u_{ti}$ and set*

$$|\tilde{u}_{min}| = \min_t |\tilde{u}_t| \quad and \quad |\tilde{u}_{max}| = \max_t |\tilde{u}_t|. \tag{65}$$

*Then for $p = \frac{1}{4}T$,*

$$\frac{\|R'\|_{HS}^2}{p} - \frac{\|R\|_{HS}^2}{T} \geq c\left(\frac{|\tilde{u}_{min}|}{n\,|u_{max}|}\right)\frac{\|R\|_{HS}^2}{T}. \tag{66}$$

The general idea behind the proof of Proposition 10 is to show that for $n = 1$, the ratio $\left(1 - \frac{p\|R\|_{HS}^2}{T\|R'\|_{HS}^2}\right)^{-1}$ can be controlled. This is done in Lemmas 11 and 12 below. In particular, Lemma 11 is a general statement about integrals of monotone real functions under certain order constraints, and Lemma 12 provides a relation of the spectrum of $R$ to that of $S'^{-1}$. It is then shown that for arbitrary $n$, $O_u S$ contains a certain copy of an $n = 1$-dimensional operator with parameters $\tilde{u}_t$, which implies the bounds.

For the case $n = 1$, stated in Lemma 12, the argument consists of showing that the spectrum of $R$ is upper and lower bounded by appropriately decaying functions, and therefore can not be "too constant". We first obtain general estimates for the integrals of such upper and lower bounded functions in the following Lemma:

**Lemma 11.** *Let $f : [0,1] \to \mathbb{R}$ be a monotone non-increasing function such that for all $x \in [0,1]$,*

$$(1-x)^2 \leq f(x) \leq M(1-x)^2, \tag{67}$$

*for some $M \geq 1$. Set $t_0 = \frac{1}{4}$. Then*

$$r(f) := \frac{\frac{1}{t_0}\int_0^{t_0} f(x)dx}{\int_0^1 f(x)dx} \geq 1 + \frac{c}{\sqrt{M}}. \tag{68}$$

*Proof.* Denote

$$I(a,b,f) = \int_a^b f(x)dx. \tag{69}$$

Write

$$r(f) = \frac{t_0^{-1} I(0, t_0, f)}{I(0, t_0, f) + I(t_0, 1, f)} = \frac{t_0^{-1}}{1 + \frac{I(t_0, 1, f)}{I(0, t_0, f)}} \tag{70}$$

and set $v := f(t_0)$. Then, among all $f$ that satisfy (67) and $f(t_0) = v$, $r(f)$ is minimized on $f$ with maximal $I(t_0, 1, f)$ and minimal $I(0, t_0, f)$. Due to the form of the constraint (67) and monotonicity, this minimizer is given by

$$\tilde{f}_v(x) = \begin{cases} (1 - x)^2 & x \in [0, t_v^-] \\ v & x \in [t_v^-, t_v^+] \\ M(1 - x)^2 & x \in [t_v^+, 1], \end{cases} \tag{71}$$

where $t_v^- := \max\{0, 1 - \sqrt{v}\}$ and $t_v^+ := 1 - \sqrt{\frac{v}{M}}$. Our problem is therefore now reduced from a minimization of $r(f)$ over the function space to a problem of minimizing $r(v) := r(\tilde{f}_v)$ over a single scalar variable $v$.

To this end, first note that we can assume w.l.o.g that $M \geq (1 - t_0)^{-2}$. Indeed, for larger $M$, the lower bound on $r(f)$ can only become smaller, since $r(f)$ would be minimized over a larger set. By construction, the value $v$ must satisfy $(1 - t_0)^2 \leq v \leq M(1 - t_0)^2$, and for $M \geq (1 - t_0)^{-2}$, the value $v = 1$ satisfies these inequalities.

Next, by direct computation (taking the derivative in $v$) one verifies that for any $M \geq (1 - t_0)^{-2}$, $r(v)$ as a function of $v$ is minimized at $v = 1$. It remains to observe that $r(1) = 1 - \frac{1}{3\sqrt{M}}$, which yields the statement of the Lemma. $\qquad \square$

We can now treat the $n = 1$ case.

**Lemma 12.** *Consider the case $n = 1$, and $p = \frac{1}{4}T$. Then*

$$\frac{\|R'\|_{HS}^2}{p} - \frac{\|R\|_{HS}^2}{T} \geq c \left( \frac{|u_{min}|}{|u_{max}|} \right) \frac{\|R\|_{HS}^2}{T}. \tag{72}$$

*Proof.* For any two operators $A, B$ we have

$$\lambda_i(AB) \leq \lambda_i(A) \|B\|_{op} \quad \text{and} \quad \lambda_i(BA) \leq \lambda_i(A) \|B\|_{op}, \tag{73}$$

see Bhatia (1997); Gohberg and Krein (1969). Note that in the case $n = 1$ and $|u_{min}| > 0$, $O_u S$ is invertible. Thus the singular values of $O_u S'$ satisfy

$$|u_{min}| \lambda_i(S') \leq \lambda_i(O_u S') \leq |u_{max}| \lambda_i(S'), \tag{74}$$

where the first inequality follows by applying (73) to $(O_u S')^{-1}$, and the second by considering $O_u S'$ itself. Equivalently, for $i = 1, \ldots, T$

$$|u_{max}|^{-1} \lambda_{T+1-i}(S'^{-1}) \leq \lambda_{T+1-i}(R) \leq |u_{min}|^{-1} \lambda_{T+1-i}(S'^{-1}), \tag{75}$$

and using Lemma 6,

$$c \frac{|u_{max}|^{-1} i}{T + 1} \leq \lambda_{T+1-i}(R) \leq |u_{min}|^{-1} \frac{\pi |u_{min}|^{-1} i}{T + 1}. \tag{76}$$

By changing the index and taking squares, we have

$$c |u_{max}|^{-2} \left( 1 - \frac{i}{T + 1} \right)^2 \leq \lambda_i^2(R) \leq c' |u_{min}|^{-2} \left( 1 - \frac{i}{T + 1} \right)^2. \tag{77}$$

Now, using an elementary discretization argument, for $p = \frac{1}{4}T$ by Lemma 11 it follows that

$$\frac{\|R'\|_{HS}^2 / p}{\|R\|_{HS}^2 / T} \geq 1 + c \frac{|u_{max}|}{|u_{min}|}, \tag{78}$$

which implies (72). $\qquad \square$

Finally, we prove Proposition 10.

*Proof.* As discussed earlier, the key to a statement such as (66) is to show that the spectrum of $R$ is non constant, and in particular has enough *small* singular values. This is equivalent to providing appropriate lower bounds on the spectrum of $O_u S$. Here we derive such bounds by comparison with an $n = 1$ case as follows: Let $V \subset \mathbb{R}^{Tn}$ be a $T$ dimensional subspace spanned by vectors for which for every time $t$, all coordinates at time $t$ are identical. Formally, for $x \in V$, for every $t$ we require that $x_{(t-1)n+i} = x_{(t-1)n+j}$ for all $i, j \leq n$.

Observe that the restriction of $O_u S$ to $V$, the operator $O_u S P_V$, acts equivalently to the $n = 1$ case operator defined by $O_{\tilde{u}} S'$. In particular, similarly to the argument in Lemma 12, it follows that

$$|\tilde{u}_{min}| \lambda_i(S') \leq \lambda_i(O_{\tilde{u}} S'). \tag{79}$$

Moreover, note that $(O_u S P_V)^* O_u S P_V = P_V S^* O_u^* O_u S P_V$ and $(O_u S)^* O_u S$ are non-negative operators and $(O_u S)^* O_u S \geq (O_u S P_V)^* O_u S P_V$ (that is, $(O_u S)^* O_u S - (O_u S P_V)^* O_u S P_V$ is non-negative). It follows that for all $i \leq T$,

$$\lambda_i(O_u S) \geq \lambda_i(O_u S P_V). \tag{80}$$

The upper bounds on the spectrum $\lambda_i(O_u S)$ may again be obtained via (73). Indeed,

$$\lambda_i(O_u S) \leq |u_{max}| \lambda_i(S) \leq |u_{max}| \lambda_{\lceil \frac{i}{n} \rceil}(S'), \tag{81}$$

where the first inequality follows from (73) while the second is due to the multiplicity $n$ of each singular value of $S'$ in $S$. The rest of the argument proceeds as in Lemma 12. $\square$

## H   Missing Values in STVE

We now discuss the treatment of missing values in STVE. Recall that our starting point is the vector form of the system, (7), which we rewrite here:

$$Y = O_u S h + z, \tag{82}$$

where $h \in \mathbb{R}^{Tn}$ and $z \in \mathbb{R}^T$ are sub Gaussian vectors and $Y = (y_1, \ldots, y_T) \in \mathbb{R}^T$ is the observation vector. Suppose that $M$ out of $T$ observation values are missing, at times $t_1, \ldots, t_M$. Set $T' = T - M$ and define a projection operator $P_A : \mathbb{R}^T \to \mathbb{R}^T$ as the operator that omits the coordinates $t_1, \ldots, t_M$. Formally, let $e(t)$, $t \leq T$, be the standard basis vector in $\mathbb{R}^T$, with 1 at coordinate $t$ and zeros elsewhere. Then

$$P_A(e(t)) = \begin{cases} 0 & \text{if } t \in \{t_1, \ldots, t_M\}, \\ e(t) & \text{otherwise.} \end{cases} \tag{83}$$

We can then rewrite (82) as

$$P_A Y = P_A O_u S h + P_A z. \tag{84}$$

Note that the vector $P_A Y$ contains only available, non-missing values of $y_t$. Similarly to the case with no missing values, we define $R$ as the Moore-Penrose inverse of $P_A O_u S$. Then we have

$$R P_A Y = R P_A O_u S h + R P_A z. \tag{85}$$

Note that since a Moore-Penrose inverse of $R$ only acts on the image of $R$, we have $R P_A = R$ and therefore

$$RY = R O_u S h + R z \tag{86}$$

similarly to the case with no missing values. The only difference here will be that $R$ will now have $T'$ rather than $T$ non-zero singular values. $R'$ will be defined similarly. The analysis of the spectrum of $R$ concerns only non-zero singular values of $R$ and holds with no change other than that $T$ should be replaced by $T'$. Consequently, the whole approach works identically with $T$ replaced by $T'$ everywhere, and in the bounds of Theorems 1 and 3 in particular.

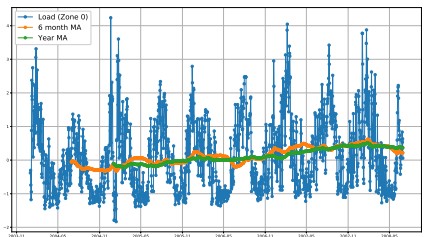
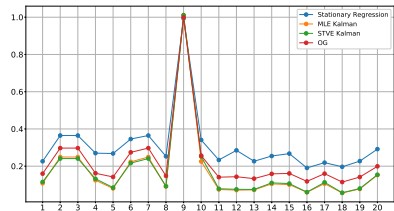

(a) Load over time, with 6 month (orange) and year (green) moving average smoothing.

(b) Total average prediction error, for each zone.

Figure 4: Supplementary Experiment Figures

# I Electricity Consumption and Temperatures Data

The raw data in Hong et al. (2014) contains electricity consumption levels for 20 nearby areas, also referred to as zones. Each zone had slightly different consumption characteristics. All figures in Section 6.2 refer to Zone 1. However, the results are similar for all zones. In particular, while Figure 2c shows the (smoothed) prediction errors of each method over time, the total error on the test set, normalized by the number of days, for each method and each zone, is shown in Figure 4b. As with zone 1, adaptive methods perform better for all zones. STVE based estimator is better than MLE in half the cases, but the differences between STVE and MLE performance are negligible compared to errors in the stationary regression or OG.

In the rest of this section we discuss the preprocessing of the data. The raw data contains hourly consumption levels for the 20 zones, and hourly temperatures from 11 weather stations with unspecified locations in the region. Here we are only interested in the total daily consumption – the total consumption over all hours of the day – for each zone. For each station we also consider the average daily temperature for the day. Moreover, since the average daily temperatures at different stations are strongly correlated ($\rho \geq 0.97$ for all pairs of stations), we only consider a single daily number, $v_t$ – an average temperature across all hours of the day and all stations.

From the total of about 234 weeks in the data, 9 non-consecutive weeks have missing loads (consumption) data, 4 in the first half of the period (train set), and 5 in the second half. For the purposes of training the stationary regression, we excluded the data points with missing values from the train set. The STVE, MLE and OG methods were trained with missing values. The prediction results of all methods were evaluated only at the points where actual values were known.

The temperatures and all the loads are normalized to have zero-mean and standard deviation 1 on the first half of the data (train set).

The MLE estimates of the parameters $\sigma^2, \eta^2$ were obtained using the DLM package of the R environment (Petris, 2010). The package uses an L-BGFS-B algorithm with numerically approximated derivatives as the underlying optimization procedure.

One immediate reason for the dependence of the load on temperature to change with time (as shown in Figure 2b for Zone 1) is that the load simply grows with time (assuming the temperatures do not exhibit a trend of the same magnitude). Indeed, it is easy to verify that there is an upward trend in the overall load, as shown in Figure 4a. However, the upward trend is non-uniform in temperature, since the parabolas in Figure 2b are not a shift of each other by a constant.

Finally, in Figure 4b average prediction errors are shown for each method and for each zone in the dataset. Zone 9 is known to be an industrial zone, (Hong, 2016), where the load does not significantly depend on the temperature, hence the higher error rates for all the methods. For the other zones the situation is similar to Zone 1, with STVE and MLE performing comparably, and better than either OG or the Stationary regression.