# OpenReview forum: "Finite Sample Analysis Of Dynamic Regression Parameter Learning"
_NeurIPS.cc/2022/Conference — NeurIPS 2022 Accept_

### Official Review · Reviewer_Wf8E · 2022-07-09

**Rating:** 6
**Confidence:** 4
**Soundness:** 3 good
**Presentation:** 2 fair
**Contribution:** 3 good

**Summary:**

The paper considers considers non-stationary linear regression with sub-Gaussian observation noise z (of variance \eta^2), where the regression vector X undergoes at each time t an additive modification by a process-noise vector h, also sub-Gaussian and zero mean with independent components of variance \sigma^2.
Estimation of the true trajectory and forecasting can be done by Kalman-filtering, once the true noise-parameters are known, and the principal contribution of the paper is to provide and analyze an algorithm to estimate the variances \sigma^2 and  \eta^2. The approach is an alternative to maximum likelihood estimation (MLE).

A compressed form of the process equations leads to a linear equation in  \sigma^2 and  \eta^2 modulo a fluctuation term. Applications of the Hanson-Wright inequalitiy bound these fluctuations and show that they are of smaller order than the coefficients in the equation. A second equally structured equation is obtained from a truncated time-series, and the two equations are solved for \sigma^2 and  \eta^2. The paper is careful to demonstrate that this system of equations is in general well conditioned relative to the magnitude of fluctuations.

An experimental section compares the algorithm to competing methods and works about well as MLE.

**Questions:**

Some typos:

l 146, square missing in expectation

l 185 and l 186, "the" missing before "sum" and "identity"

l 199, I believe it should be \gamma instead of \lambda

p 19, equation (78). u_max and u_min seem exchanged.

**Limitations:**

There is no section on limitations, but I am also not clear on what it should contain for this paper.

**Strengths And Weaknesses:**

Strengths:
1. In contrast to MLE the approach has performance guarantees for the more general sub-Gaussian noise model.
2. The paper contains a number of interesting mathematical tricks.

Weaknesses:
1. The presentation is a bit opaque. Section 4 could perhaps be a lot more readable if portioned into sub-sections: Obtaining the representation (7) - why using the pseudo-inverse and obtaining (11) - obtaining the second equation (12) - etc.
2. The role of Theorem 3 and Section G could be better explained. The most problematic part of the error bounds in Corollary 2 (inverse parenthesis) can already be observed when executing the algorithm, so from a practical point of view Theorem 3 and Section G are superfluous. Their interest lies in showing that generically the division in step 4 of the algorithm is unproblematic.

---

> ### Author Response · Authors · 2022-08-01
> **author feedback**
>
> Many thanks for the favourable review! We would like especially to thank the reviewer for considering the material worth the attention enough to spot a typo as deep in the paper as Eq. (78) on page 19! This is greatly encouraging.
>
>
> The typos will be fixed and the presentation comments will be taken into account.
> As mentioned in the review, indeed the role of Theorem 3 and Section G are to provide indication that generally the system (11)-(12) is well conditioned.
>
>
> Since a significant part the mathematical arguments in this paper
> are in Theorem 3, we felt it necessary to present it in the main text.
> One aspect of Theorem 3, that was not mentioned in the text at all, is that the upper bound in (20) in Theorem 3 is required to prove Theorem 1.  The other aspect, (mentioned on lines 205-209, admittedly briefly ) is that it implies that both coefficients of eq. (11) are non-negligible, providing at least some indication that both $\sigma$ and $\eta$ should be recoverable.  We will try to clarify the presentation.

---

### Official Review · Reviewer_1GcG · 2022-07-11

**Rating:** 7
**Confidence:** 4
**Soundness:** 4 excellent
**Presentation:** 4 excellent
**Contribution:** 3 good

**Summary:**

This paper provides novel finite-sample guarantees for parameter estimation in the dynamic linear regression setting, where the true linear model’s coefficients vary across time. By recharacterizing the dynamic regression problem into a linear dynamical system, the authors note that existing methods, such as the Kalman Filter, rely on knowing certain variance parameters. While one could estimate these variance parameters using maximum likelihood estimation, the accompanying Gaussian noise assumption is perhaps too strong and there are only asymptotic guarantees of the validity of the estimates. The authors of this paper have developed a procedure with finite sample guarantees for estimating the variance parameters with a less onerous sub-Gaussian noise assumption for the dynamical system. With a simulation, the authors demonstrate that the novel estimator’s error decays at a rate in line with the theoretical guarantees. In an empirical example of predicting electricity consumption over time, the authors show that their estimator, once inputted into a Kalman Filter algorithm, performs comparably to using maximum likelihood estimations as the inputs.

**Questions:**

The field of data assimilation often deals with methods like the Kalman filter with unknown parameters; I was surprised not to see this mentioned and it would be helpful for the authors to explicitly address their contributions relative to work in that field. For instance, how does the submission compare with https://arxiv.org/abs/1903.09122, https://arxiv.org/abs/1912.12309, or https://arxiv.org/abs/2001.06270?

**Limitations:**

The paper discusses limitations naturally for the most part, given the extensive discussion of comparable methods and highlighting the new assumptions the novel approach relies on. The authors do note, briefly, at the end of the Experiments section that their novel approach performs “practically identical” to the maximum-likelihood estimation approach. While this is not at odds with the main claim of the authors of finally providing an estimator with finite sample guarantees, they are essentially acknowledging a limitation of the new estimator’s performance versus existing methods.

As far as the discussion on societal impact and the like, the paper eschews such discussion due to the theoretical nature of research.


**Strengths And Weaknesses:**

Strengths

The paper provides novel finite-sample guarantees in a useful generalization of linear regression. The proofs seem innovative and non-trivial, providing bounds that may have previously seemed out-of-reach. In particular, the use of existing theory on the discrete Laplacian operator on the line seems like an especially novel approach that other theoreticians will find intriguing and useful.

The authors do a great job of showing how the problem set-up, which starts very general, can be recharacterized into a more specific problem that has been well-studied. No less, the authors home in on an aspect that has previously been untouched, providing bounds on estimating variance parameters in the dynamical system.

The clarity of the paper’s writing and the cogent overview of the literature were also appreciated. Via the problem set-up section, the authors provided very clear exposition on the links to previous literature and what the novel contributions of this paper is, aided by uniting the notation of their approach with others. The writing was clear throughout, and the authors provided ample details and justification for the proof sketches in the body of the paper.

Weaknesses

The two main ways to improve the paper are in enriching the experimental section and making more clear the importance of the novel finite sample guarantees.

In the empirical sections, it seems like MLE often does not differ much from the novel approach. While the authors briefly note this, it would be useful to have them discuss the implications of this more. For example, under what circumstances do they differ, and can we learn anything from those settings? In addition, for the performance comparisons on the real world data example, it would be helpful to list MSE or some other test metrics. While plotting the predictions versus the truth was certainly a step in the right direction, it could seem like obfuscation to not provide the metrics, especially since it seems like the novel estimator may have performed worse than the MLE. For the simulation study, the sensitivity analysis provided on the number of samples was certainly of utmost importance. However, the authors should also note why they chose some of the settings (e.g., why the number of covariates was five or the choices for the ground truth variance parameters). It was not clear whether these settings were chosen due to computational constraints, to enhance the signal-to-noise ratio, to adhere to some realistic empirical examples, or if they were just drawn out of a hat.

It may be useful to provide more color on the usefulness of better variance parameter estimates and these guarantees. Are the parameters only useful insofar as inputs to the Kalman Filter and related approaches to get better dynamic coefficient estimates? Or are these variance parameters intrinsically interesting?

---

> ### Author Response · Authors · 2022-08-01
> **author feedback**
>
> Thank you very much for the favourable review, encouraging comments, and the references!
>
>
> >It may be useful to provide more color on the usefulness of better variance parameter estimates and these guarantees. Are the parameters only useful insofar as inputs to the Kalman Filter and related approaches to get better dynamic coefficient estimates? Or are these variance parameters intrinsically interesting?
>
> One possible interpretation of the variance parameters is as the speed of change of the regression vector, sometimes also referred to as the amount of memory of the system. The idea is that if the process noise $\sigma^2$ is large, or the observation noise $\eta^2$ is small, then the regression vector $X_t$ will be determined only by the last few observations. If, on the other hand, $\sigma^2$ is small and $\eta^2$ is large, then each new observation will have only small influence on the state $X_t$, i.e. a ``long memory" process.  In the extreme case of $\sigma^2 = 0$, we recover the standard non-dynamic regression where $X_t$ does not change. However, admittedly, the difference of the roles of $\sigma^2$ and $\eta^2$ in this interpretation would be more challenging to describe consciously. Some additional discussion along these lines, along with an analogy to learning rates in online learning, is in the paper on lines 44-59.
>
>
> **References**
>
> We now compare the results of the paper with the works provided in the review.  All the references will be added to the paper.
>
> _Regarding the paper_  https://arxiv.org/abs/2001.06270:
>
> Thank you for pointing us to this reference.
> From the point of view of our paper, to the best of our understanding, the results as in reference above constitute a far reaching extension of the standard maximum-likelihood or Bayesian estimators for linear dynamical system. The goal of these works is to model complex situations, involving non-linear dynamics expressed with neural networks. The optimization is typically done via EM or gradient based algorithms.
>
> Due to the complexity and generality of the situation, these methods typically come with no theoretical guarantees. The goal in our paper (and other work, such as the references below) is to provide theoretical guarantees, which, at least at the moment, entails considering a much simpler situation.
>
>
>
> _Regarding  the references_ https://arxiv.org/abs/1903.09122 and https://arxiv.org/abs/1912.12309:
>
> These papers analyze the following
> LDS:
>
>
> $X_{t+1}  = A X_t + h_t $
>
> $Y_{t} = C X_{t} + z_t$
>
> where $A,C$ are unknown dynamics and observation operators. On the other hand, here we are interested in the system
>
> $X_{t+1}  =  X_t + h_t $
>
> $Y_{t} = <X_{t},u_t> + z_t$
>
>
> In the former system the dynamics $A$ may be more general, and observation operator $C$ is apriori unknown but constant. On the other hand, in our version the observation operators $u_t$ are known but *change  with time*. This is crucial for the dynamic regression application, since $u_t$ are the feature vectors provided by user.
>
> Technically, if $C$ is assumed to be constant, one can use stationary state arguments
> to recover the parameters. However, for general sequence $u_t$ the Kalman gain does not need to converge, implying that there is no stationary state and that such arguments are not possible. We take a different approach by deriving spectral bounds for $O_u S$ for an arbitrary sequence $u_t$. Additional related  discussion of this matter can be found on lines 100-108.
>
> This comment also appears in our discussion in review p7x7, where references to a similar line of work (i.e., the first system above) were provided.

---

> > ### Comment · Reviewer_1GcG · 2022-08-08
> > **I have read the rebuttal**
> >
> > I have read the rebuttal. I'm disappointed that the authors did not address my questions and concerns about the experiments, but the answers to questions about connections to data assimilation were adequate. I would have preferred to see some promise from the authors to include this highly relevant section of the literature in a revised version of the paper, particularly since two reviewers had the same concern independently.
> >
> > I have increased my score in light of my improved understanding of the novelty of the analysis.

---

> > > ### Author Response · Authors · 2022-08-09
> > > **author response**
> > >
> > > Thanks so much for the comment and for raising the score!
> > >
> > > Regarding the references, we will certainly include all the references in this and other reviews in the final version of the paper. Apologies if we have not clarified this in the earlier response.
> > >
> > >
> > > Regarding the experiments: Unfortunately, we believe we have not seen part of the current experiments comments earlier. Is it possible that they were added via edits after the bulk of the review was in place? (We are not notified about edits). Of course it might also have been an oversight on our part. We'll try to answer this now.
> > >
> > > **a)**
> > > >under what circumstances do they (MLE and STVE) differ, and can we learn anything from those settings?
> > >
> > > One possible setting where the methods might differ, and STVE may have an advantage, is
> > > when the noise is non-Gaussian. This is due to the fact that MLE estimates maximize the *Gaussian* likelihood. This setting was also suggested in Review p7x7.   We plan to perform such an evaluation and add it to the paper.
> > >
> > > **b)** MSE and additional metrics will be added in the final version of the paper.
> > >
> > > **c)** With synthetic data, our goal was to demonstrate the $1/\sqrt{T}$ decay of the error.
> > > The dimension $n$ was chosen to be similar to the electricity example, and we wanted neither  of the $\sigma$ or $\eta$ parameters to be too small compared to the other (this would have made the problem simpler, empirically, in our experience). We will add an evaluation over a range of parameters.

---

### Official Review · Reviewer_p7x7 · 2022-07-12

**Rating:** 7
**Confidence:** 3
**Soundness:** 4 excellent
**Presentation:** 3 good
**Contribution:** 3 good

**Summary:**

This paper studies estimation of process and observation noise variance in a subclass of linear dynamical systems whose states evolve according to $X_{t+1} = X_t + h_t$ and observations follow $Y_{t} = \langle X_t, u_t \rangle + z_t$, in which $u_t$ are known observation vectors. The paper presents the first algorithm for the estimation of variance in this class of dynamical systems that enjoys finite-sample guarantees. The approach revolves around forming variance estimators that exploit data-dependent operators.

**Questions:**

Additional related work suggestions that seem relevant:
- Rashidiejad, Paria, Jiantao Jiao, and Stuart Russell. "SLIP: Learning to Predict in Unknown Dynamical Systems with Long-Term Memory." 34th Conference on Neural Information Processing Systems (NeurIPS 2020), Vancouver, Canada. 2020.
- Tsiamis, Anastasios, and George J. Pappas. "Linear systems can be hard to learn." In 2021 60th IEEE Conference on Decision and Control (CDC), pp. 2903-2910. IEEE, 2021.

**Limitations:**

Limitations are discussed. The work is mainly theoretical.

**Strengths And Weaknesses:**

**Strengths**
- The paper presents the first variance estimation for the LDS subclass with finite-sample guarantees. The resulting algorithm is simple.
- As far as I know, techniques used in the derivation of the approach are original.
- The paper is reasonably well-written. The authors provide intuition regarding their choices and approach and compare them with alternatives.

**Weaknesses**
- The system considered in this work is a subclass of LDS with scalar observations, which can limit the applicability of the approach.
- The experiment section is a bit weak (although the paper is mainly theoretical). At least based on the experiment in Section 6.2, STVE does not perform better than MLE (followed by the Kalman filter). Additional empirical evaluation that highlights the benefits of STVE (e.g. in handling non-Gaussian noise) would strengthen the paper).

---

> ### Author Response · Authors · 2022-08-01
> **author feedback**
>
> Thank you very much for the favourable review and the references!
>
> The references will be added to the paper.
>
>
>
>
> Just in case, we would like to clarify the relation between our results and the results in the references. The papers in the references analyze the following
> LDS:
>
>
> $X_{t+1}  = A X_t + h_t $
>
> $Y_{t} = C X_{t} + z_t$
>
> where $A,C$ are unknown dynamics and observation operators. On the other hand, here we are interested in the system
>
> $X_{t+1}  =  X_t + h_t $
>
> $Y_{t} = <X_{t},u_t> + z_t$
>
>
> In the former system the dynamics $A$ may be more general, and observation operator $C$ is apriori unknown but constant. On the other hand, in our version the observation operators $u_t$ are known but *change  with time*. This is crucial for the dynamic regression application, since $u_t$ are the feature vectors provided by user.
>
> Technically, if $C$ is assumed to be constant, one can use stationary state arguments
> to recover the parameters. However, for general sequence $u_t$ the Kalman gain does not need to converge, implying that there is no stationary state and that such arguments are not possible. We take a different approach by deriving spectral bounds for $O_u S$ for an arbitrary sequence $u_t$. Additional related  discussion of this matter can be found on lines 100-108.
>
>
> This comment also appears in our discussion in review 1GcG, where references to a similar line of work (i.e. the first system above) were provided.

---

> > ### Comment · Reviewer_p7x7 · 2022-08-09
> > **Thank you for your response**
> >
> > I thank the authors for providing a detailed comparison with the LDS line of work and clarifying the technical difficulties that arise in analyzing the system with time-varying observation operators. I stick to my score in favor of accepting this paper.

---

### Meta-Review · Area_Chair_RVka · 2022-08-26

**Recommendation:** Accept
**Confidence:** Certain

**Metareview:**

The paper considers the estimation of process and observation noise variances in a subclass of linear dynamical systems and provides algorithms with finite sample guarantees. The math is novel and the results are interesting. I am happy to recommend acceptance.


**Award:**

No

---

### Decision · Program_Chairs · 2022-09-14

Accept